# Combining Symbolic Expressions and Black-box Function Evaluations in Neural Programs

**Forough Arabshahi**
University of California
Irvine, CA
farabsha@uci.edu

**Sameer Singh**
University of California
Irvine, CA
sameer@uci.edu

**Animashree Anandkumar**
California Institute of Technology
Pasadena, CA
anima@caltech.edu

## Abstract

Neural programming involves training neural networks to learn programs, mathematics, or logic from data. Previous works have failed to achieve good generalization performance, especially on problems and programs with high complexity or on large domains. This is because they mostly rely either on black-box function evaluations that do not capture the structure of the program, or on detailed execution traces that are expensive to obtain, and hence the training data has poor coverage of the domain under consideration. We present a novel framework that utilizes black-box function evaluations, in conjunction with symbolic expressions that define relationships between the given functions. We employ tree LSTMs to incorporate the structure of the symbolic expression trees. We use tree encoding for numbers present in function evaluation data, based on their decimal representation. We present an evaluation benchmark for this task to demonstrate our proposed model combines symbolic reasoning and function evaluation in a fruitful manner, obtaining high accuracies in our experiments. Our framework generalizes significantly better to expressions of higher depth and is able to fill partial equations with valid completions.

## 1 Introduction

Human beings possess impressive abilities for abstract mathematical and logical thinking. It has long been the dream of computer scientists to design machines with such capabilities: machines that can automatically learn and reason, thereby removing the need to manually program them. Neural programming, where neural networks are used to learn programs, mathematics, or logic, has recently shown promise towards this goal. Examples of neural programming include neural theorem provers, neural Turing machines, and neural program inducers, e.g. Loos et al. (2017); Graves et al. (2014); Neelakantan et al. (2015); Bošnjak et al. (2017); Allamanis et al. (2017). They aim to solve tasks such as learning functions in logic, mathematics, or computer programs (e.g. logical or, addition, and sorting), prove theorems and synthesize programs.

Most works on neural programming either rely only on black-box function evaluations (Graves et al., 2014; Balog et al., 2017) or on the availability of detailed program execution traces, where entire program runs are recorded under different input conditions (Reed & De Freitas, 2016; Cai et al., 2017). Black-box function evaluations are easy to obtain since we only need to generate inputs and outputs to various functions in the domain. However, by themselves, they do not result in powerful generalizable models, since they do not have sufficient information about the underlying structure of the domain. On the other hand, execution traces capture the underlying structure, but, are generally harder to obtain under many different input conditions; even if they are available, the computational complexity of incorporating them is significant. Due to the lack of good coverage, these approaches fail to generalize to programs of higher complexity and to domains with a large number of functions. Moreover, the performance of these frameworks is severely dependent on the nature of execution traces: more efficient programs lead to a drastic improvement in performance (Cai et al., 2017), but such programs may not be readily available.

In many problem domains, in addition to function evaluations, one typically has access to more information such as symbolic representations that encode the relationships between the given variables

and functions in a succinct manner. For instance, in physical systems such as fluid dynamics or robotics, the physical model of the world imposes constraints on the values that different variables can take. Mathematics and logic are other domains in which expressions are inherently symbolic. In the domain of programming languages, declarative languages explicitly declare variables in the program. For instance, database query languages (e.g., SQL), regular expressions, and functional programming. Declarative programs greatly simplify parallel programs through the generation of symbolic computation graphs, and have thus been used in modern deep learning packages, such as Theano, TensorFlow, and MxNet. Therefore, rich symbolic expression data is available for many domains. We will show in this paper, that incorporating this type of information, as well as black-box function evaluations, will result in models that are more generalizable.

**Summary of Results:** We introduce a flexible and a scalable neural programming framework that combines the knowledge of symbolic expressions with black-box function evaluations. To our knowledge, we are the first to consider such a combined framework. We demonstrate that this approach outperforms existing methods by a significant margin, using only a small amount of training data. The paper has three main contributions. (1) We design a neural architecture to incorporate both symbolic expressions and black-box function evaluation data. (2) We evaluate it on tasks such as equation verification and completion in the domain of mathematical equation modeling. (3) We propose a data generation strategy for both symbolic expressions and black-box function evaluations that results in good balance and coverage.

We consider learning mathematical equations and functions as a case study, since it has been used extensively in previous neural programming works, e.g. Zaremba et al. (2014); Allamanis et al. (2017); Loos et al. (2017). We employ tree LSTMs to incorporate the symbolic expression tree, with one LSTM cell for each mathematical function. The parameters of the LSTM cells are shared across different expressions, wherever the same function is used. This weight sharing allows us to learn a large number of mathematical functions simultaneously, whereas most previous works aim at learning only one or few mathematical functions. We then extend tree LSTMs to not only accept symbolic expression input, but also numerical data from black-box function evaluations. We employ tree encoding for numbers that appear in function evaluations, based on their decimal representation (see Fig. 1c). This allows our model to generalize to unseen numbers, which has been a struggle for neural programing researchers so far. We show that such a recursive neural architecture is able to generalize to unseen numbers as well as to unseen symbolic expressions.

We evaluate our framework on two tasks: equation verification and completion. Under equation verification, we further consider two sub-categories: verifying the correctness of a given symbolic identity as a whole, or verifying evaluations of symbolic expressions under given numerical inputs. Equation completion involves predicting the missing entry in a mathematical equation. This is employed in applications such as mathematical question answering (QA). We establish that our framework outperforms existing approaches on these tasks by a significant margin, especially in terms of generalization to equations of higher depth and on domains with a large number of functions.

We propose a novel dataset generation strategy to obtain a balanced dataset of correct and incorrect symbolic mathematical expressions and their numerical function evaluations. Previous methods do an exhaustive search of all possible parse trees and are therefore, limited to symbolic trees of small depth (Allamanis et al., 2017). Our dataset generation strategy relies on dictionary look-up and sub-tree matching and can be applied to any domain by providing a basic set of axioms as inputs. Our generated dataset has good coverage of the domain and is key to obtaining superior generalization performance. We are also able to scale up our coverage to include about $3.5\times$ mathematical functions compared to the previous works (Allamanis et al., 2017; Zaremba et al., 2014).

**Related work:** Early work on automated programming used first order logic in computer algebra systems such as Wolfram Mathematica and Sympy. However, these rule-based systems required extensive manual input and could not be generalized to new programs. Graves et al. (2014) introduced using memory in neural networks for learning functions such as grade-school addition and sorting. Since then, many works have extended it to tasks such as program synthesis, program induction and automatic differentiation (Bošnjak et al., 2017; Tran et al., 2017; Balog et al., 2017; Parisotto et al., 2017; Reed & De Freitas, 2016; Mudigonda et al., 2017; Sajovic & Vuk, 2016; Piech et al., 2015).

Based on the type of data that is used to train the models, frameworks in neural programming are categorized under 4 different classes. (1) Models that use black-box function evaluation data

(Graves et al., 2014; Balog et al., 2017; Parisotto et al., 2017; Zaremba et al., 2014), (2) models that use program execution traces (Reed & De Freitas, 2016; Cai et al., 2017). (3) models that use a combination of black-box input-output data and weak supervision from program sketches (Bošnjak et al., 2017; Neelakantan et al., 2015) and finally (4) models that use symbolic data (Allamanis et al., 2017; Loos et al., 2017). Our work is an extension of models of category 3 which uses symbolic data instead of weak supervision. As stated in Section 1, an example of symbolic data is the computation graph of a program which is different from program execution traces used in models of category 2 such as (Reed & De Freitas, 2016). These high-level symbolic expressions summarize the behavior of the functions in the domain and apply to many groundings of different inputs as opposed to the Neural Programmer-Interpreter (Reed & De Freitas, 2016). Therefore, we can obtain generalizable models that are capable of function evaluation. Moreover, this combination allows us to scale up the domain and model more functions as well as learn more complex structures.

One of the extensively studied applications of neural programming is reasoning with mathematical equations. These works include automated theorem provers (Loos et al., 2017; Rocktäschel & Riedel, 2016; Yuan; Alemi et al., 2016; law Chojecki, 2017; Kaliszyk et al., 2017) or computer algebra-like systems (Allamanis et al., 2017; Zaremba et al., 2014). Our work is closer to the latter, under this categorization, however, the problem that we solve is in nature different. Allamanis et al. (2017) and Zaremba et al. (2014) aim at simplifying mathematical equations by defining equivalence classes of symbolic expressions that can be used in a symbolic solver. Our problem, on the other hand, is mathematical equation verification and completion which has broader applicability, e.g. our proposed model can be used in mathematical question answering systems.

Recent advances in symbolic reasoning and natural language processing have indicated the significance of applying domain structure to the models to capture compositionality and semantics. Socher et al. (2011; 2012) proposed tree-structured neural networks for natural language parsing and neural image parsing. Cai et al. (2017) proposed using recursion for capturing the compositionality of computer programs. Both Zaremba et al. (2014) and Allamanis et al. (2017) used tree-structured neural networks for modeling mathematical equations. Tai et al. (2015) introduced tree-structured LSTM for semantic relatedness in natural language processing. We will show that this powerful model outperforms other tree-structured neural networks for validating mathematical equations.

## 2 MATHEMATIC EQUATION MODELING

We now address the problem of modeling mathematical equations. Our goal is to verify the correctness of a mathematical equation. This then enables us to perform equation completion. We limit ourselves to the domain of trigonometry and elementary algebra in this paper.

In this section, we first discuss our grammar that explains the domain under our study. we later describe how we generate a dataset of correct and incorrect symbolic equations within our grammar. We talk about how we combine this data with a few input-output examples to enable function evaluation. This dataset allows us to learn representations for the functions that capture their semantic properties, i.e. how they relate to each other, and how they transform the input when applied. We interchangeably use the word identity for referring to mathematical equations and input-output data to refer to function evaluations.

### 2.1 GRAMMAR

Let us start by defining our domain of the mathematical identities using the context-free grammar notation. Identities (I), by definition, consist of two expressions that we are trying to verify (Eq. (1)). A mathematical expression, represented by $E$ in Eq. (2), is composed either of a terminal ($T$), such as a constant or a variable, a unary function applied to any expression ($F_1$), or a binary function applied to two expression arguments ($F_2$). Without loss of generality, functions that take more than two arguments, i.e. $n$-ary functions with $n > 2$, are omitted from our task description, since $n$-ary functions like addition can be represented as the composition of multiple binary addition functions. Therefore, this grammar covers the entire space of trigonometric and elementary algebraic identities. The trigonometry grammar rules are thus as follows:

$$I \rightarrow =(E, E), \ \neq(E, E) \tag{1}$$
$$E \rightarrow T, F_1(E), F_2(E, E) \tag{2}$$

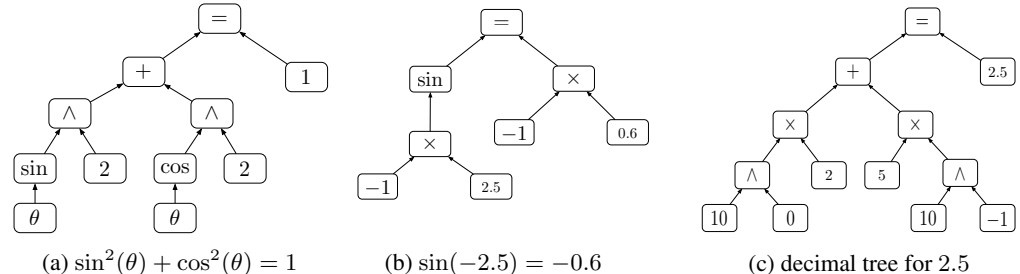

(a) $\sin^2(\theta) + \cos^2(\theta) = 1$     (b) $\sin(-2.5) = -0.6$     (c) decimal tree for 2.5

Figure 1: **Identities and their Expression Trees.** with (a) a symbolic expression, (b) a function evaluation, and (c) a number represented as the decimal tree (also part of the function evaluation data)

Table 1: Symbols in our grammar, i.e. the functions, variables, and constants

| **Unary functions, $F_1$** | | | | | **Terminal, $T$** | | **Binary, $F_2$** |
|---|---|---|---|---|---|---|---|
| sin | cos | csc | sec | tan | 0 | 1 | $+$ |
| cot | arcsin | arccos | arccsc | arcsec | 2 | 3 | $\times$ |
| arctan | arccot | sinh | cosh | csch | 4 | 10 | $\wedge$ |
| sech | tanh | coth | arsinh | arcosh | 0.5 | $-1$ | |
| arcsch | arsech | artanh | arcoth | exp | 0.4 | 0.7 | |
| | | | | | $\pi$ | $x$ | |

$$F_1 \rightarrow \sin, \cos, \tan, \ldots \tag{3}$$

$$F_2 \rightarrow +, \wedge, \times, \ldots \tag{4}$$

$$T \rightarrow -1, 0, 1, 2, \pi, x, y, \ldots, \text{any number of precision 2 in [-3.14,+3.14]} \tag{5}$$

Table 1 presents the complete list of functions and symbols as well as examples of the terminals of the grammar. Note that we exclude subtraction and division because they can be represented with addition, multiplication and power, respectively. Furthermore, the equations can have as many variables as needed.

The above formulation provides a parse tree for any symbolic and function evaluation expression, a crucial component for representing the equations in a model. Figure 1 illustrates 3 examples of an identity in our grammar in terms of its expression tree. It is worth noting that there is an implicit notion of depth of an identity in the expression tree. Since deeper equations are compositions of multiple simpler equations, validating higher-depth identities requires reasoning beyond what is required for identities with lower depths, and thus depth of an equation is somewhat indicative of the complexity of the mathematical expression. However, depth is not sufficient; some higher depth identities such as $1+1+1+1 = 4$ may be much easier to verify than $\tan^2 \theta + 1 = \mathrm{acos}^2 \theta$. Symbolic and function evaluation expressions are differentiated by the type of their terminals. Symbolic expressions have terminals of type constant or variable, whereas function evaluation expressions have constants and numbers as terminals. We will come back to this distinction in section 3 where we define our model. As shown in Table 1, our domain includes 28 functions. This scales up the domain in comparison to the state-of-the-art methods that use up to 8 mathematical functions (Allamanis et al., 2017; Zaremba et al., 2014). We will also show that our expressions are of higher complexity as we consider equalities of depth up to 4, resulting in trees of size at most 31. Compared to the state-of-the-art methods that use trees of size at most 13 (Allamanis et al., 2017).

**Axioms:** We refer to a small set of basic trigonometric and algebraic identities as axioms. These axioms are gathered from the Wikipedia page on trigonometric identities[1] as well as manually specified ones covering elementary algebra. This set consists of about 140 identities varying in depth from 1 to 7. Some examples of our axioms are (in ascending order of depth), $x = x$, $x + y = y + x$, $x \times (y \times z) = (x \times y) \times z$, $\sin^2(\theta) + \cos^2(\theta) = 1$ and $\sin(3\theta) = -4\sin^3(\theta) + 3\sin(\theta)$. These axioms represent the basic properties of the mathematical functions in trigonometry and algebra,

---

[1] https://en.wikipedia.org/wiki/List_of_trigonometric_identities

but do not directly specify their input/output behavior. The axioms, consisting of only positive (or correct) identities, serve as a starting set for generating a dataset of mathematical identities.

## 2.2 Dataset of Mathematical Equations

In order to provide a challenging and accurate benchmark for our task, we need to create a large, varied collection of correct and incorrect identities, in a manner that can be extended to other domains in mathematics easily. Our approach is based on generating new mathematical identities by performing local random changes to known identities, starting with 140 axioms described above. These changes result in identities of similar or higher complexity (equal or larger depth), which may be correct or incorrect, that are valid expressions within the grammar.

**Generating Possible Identities:** To generate a new identity, we select an equation at random from the set of known equations, and make local changes to it. In order to do this, we first randomly select a node in the expression tree, followed by randomly selecting one of the following actions to make the local change to the equation at the selected node:

- *ShrinkNode:* Replace the node, if it's not a leaf, with one of its children, chosen randomly.
- *ReplaceNode:* Replace the symbol at the node (i.e. the terminal or the function) with another compatible one, chosen randomly.
- *GrowNode:* Provide the node as input to another randomly drawn function $f$, which then replaces the node. If $f$ takes two inputs, the second input will be generated randomly from the set of terminals.
- *GrowSides:* If the selected node is an equality, either add or multiply both sides with a randomly drawn number, or take both sides to the power of a randomly drawn number.

At the end of this procedure we use a symbolic solver, sympy (Meurer et al., 2017), to separate correct equations from incorrect ones. Since we are performing the above changes randomly, the number of generated incorrect equations are overwhelmingly larger than the number of correct identities. This makes the training data highly unbalanced and is not desired. Therefore, we propose a method based on sub-tree matching to generate new correct identities.

**Generating Additional Correct Identities:** In order to generate only correct identities, we follow the same intuition as above, but only replace structure with others that are *equal*. In particular, we maintain a dictionary of valid statements (*mathDictionary*) that maps a mathematical statement to another. For example, the dictionary *key* $x + y$ has *value* $y + x$. We use this dictionary in our correct equation generation process where we look up patterns from the dictionary. More specifically, we look for keys that match a subtree of the equation then replace that subtree with the pattern of the value of the key. E.g. given input equation $\sin^2(\theta) + \cos^2(\theta) = 1$, this subtree matching might produce equality $\cos^2 \theta + sin^2(\theta) = 1$ by finding *key-value* pair $x + y : y + x$.

The initial *mathDictionaty* is constructed from the input list of axioms. At each step of the equation generation, we choose one equation at random from the list of correct equations so far, and choose a random node $n$ of this equation tree for changing. We look for a subtree rooted at $n$ that matches one or several dictionary keys. We randomly choose one of the matches and replace the subtree with the value of the key by looking up the *mathDictionary*.

We generate all possible equations at a particular depth before proceeding to a higher depth. In order to ensure this, we limit the depth of the final equation and only increase this limit if no new equations are added to the correct equations for a number of repeats. Some examples of correct and incorrect identities generated by out dataset generation method is given in Table 2.

**Generating Function Evaluation data:** We generate a few input-output examples from a specific range of numbers for the functions in our domain. For unary functions, we randomly draw floating point numbers of fixed precision in the range and evaluate the functions for the randomly drawn number. For binary functions we repeat the same with two randomly generated numbers. Note that function evaluation results in identities of depths 2 and 3.

**Generating Numerical Expression Trees:** It is important for our dataset to also have a generalizable representation of the numbers. We represent the floating point numbers with their decimal expansion

Table 2: Examples of generated equations

| Examples of correct identities | Examples of incorrect identities |
|---|---|
| $1^2 = x^{-1 \times 0}$ | $0.5^{x+2} = sin(0.5)^{x+2}$ |
| $(\arctan 10)^{2^2} = (\arctan 10)^{3+1}$ | $\pi \times \csc(x) = -\csc(x)$ |
| $x \times (-1 + x) = x \times (x - 1)$ | $-4 = -4^x$ |
| $x^1 = x + 0$ | $\frac{\sqrt{2}}{2} \times \sqrt{x} = \sqrt{x}$ |

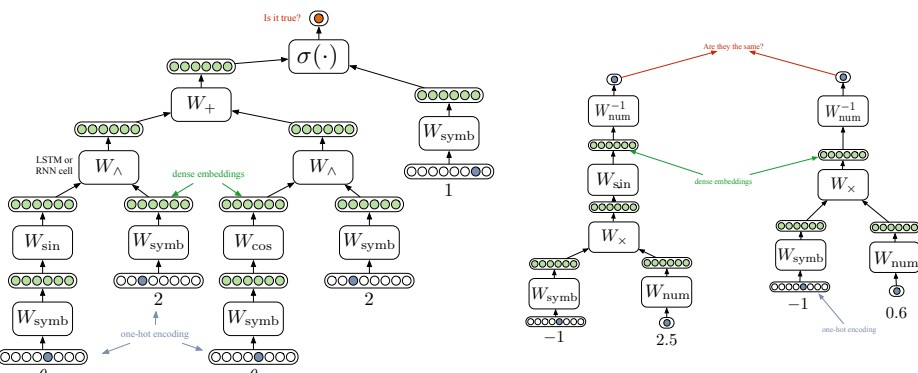

Figure 2: Tree-structured recursive neural model, for the trees in Figure 1a (left) and 1b (right)

which is representable in our grammar. In order to make this clear, consider number 2.5. In order to represent this number, we expand it into its decimal representation $2.5 = 2 \times 10^0 + 5 \times 10^{-1}$ and feed this as one of the function evaluation expressions for training (Figure 1c). Therefore, we can represent floating point numbers of finite precision using integers in the range [-1,10].

## 3  TREE LSTM ARCHITECTURE FOR MODELING EQUATIONS

Analogous to how humans learn trigonometry and elementary algebra, we propose using basic axioms to learn about the properties of mathematical functions. Moreover, we leverage the underlying structure of each mathematical identity to make predictions about their validity. Both Zaremba et al. (2014) and Allamanis et al. (2017) validate the effectiveness of using tree-structured neural networks for modeling equations. Tai et al. (2015) show that Tree LSTMs are powerful models for capturing the semantics of the data. Therefore, We use the Tree LSTM model to capture the compositionality of the equation and show that it improves the performance over simpler tree-structured, a.k.a recursive, neural networks. We describe the details of the model and training setup in this section.

**Tree LSTM Model for Symbolic Expressions and Function Evaluations**  The structure of Tree LSTM mirrors the parse-tree of each input equation. As shown in Figure 1, the input equation's parse tree is inherent in each equation. As described in section 2.1 an equation consists of terminals and binary and unary functions. Terminals are input to Tree LSTM through the leaves that embeds their representation using vectors. Each function is associated with an LSTM block with its own weights, with the weights shared among all appearances of the function in different equations. we predict the validity of each equation at the root of the tree.

The architecture of the neural network is slightly different for symbolic expressions compared to function evaluation expressions. Recall from section 2.1, that the two are distinguished by their terminal types. This directly reflects to the structure of the network used in the leaves for embedding. Moreover, we use different loss functions for each type of expression as described below.

- *Symbolic expressions* These expressions consist of constants and symbols. These terminals are represented with their one-hot encoding and are passed through *symbol*, a single layer neural network block. The validity of a symbolic expression is verified by computing the

> dot product of the left-hand-side and right-hand-side vector embeddings and applying the logistic function.
>
> • *function evaluation expressions* In order to encode the terminals of function evaluation expressions we train an autoencoder. The encoder side embeds the floating point numbers into a high-dimensional vector space. We call this a *number* block. The decoder of this auto-encoder is trained for predicting the floating point number given an input embedding. We call this the *decoder* block. We pass the output vector embedding of the left-hand-side and right-hand-side to the *decoder* block. The validity of a function evaluation is then computed by minimizing the MSE loss of the decoder outputs of each side.

Figure 2 illustrates our tree LSTM structure constructed from the parse-tree of the equations in Figures 1a and 1b. [2]

**Baseline Models** We compare our proposed model with chain-structured neural networks, such as sequential Recurrent Neural Networks (RNN), LSTMS's as well as tree-structured neural networks (TreeNN's) consisting of fully connected layers (Socher et al., 2011; Zaremba et al., 2014). It should be noted that both these papers discover equivalence classes in a dataset, and since our data consists of many equivalence classes especially for the function evaluation data, we do not use the EqNet model proposed in (Allamanis et al., 2017) as a baseline. Another baseline we have used is Sympy. Given each equality, Sympy either returns True, or False, or returns the input equality in its original form (indicating that sympy is incapable of deciding whether the equality holds or not). Let's call this the Unsure class. In the reported Sympy accuracies we have treated the Unsure class as a miss-classification. It should be noted, however, that Since Sympy is used at time of data generation to verify the correctness of the generated equations, its accuracy for predicting correct equations in our dataset is always 100%. Therefore, the degradation in Sympy's performance in Table 3 is only due to incorrect equations. It is interesting to see Sympy's performance when another oracle is used for validating correct equalities.

As we will show in the experiments, the structure of the network is crucial for equation verification and equation completion. Moreover, by adding function evaluation data to the tree-structured models we show that using this type of data not only broadens the applicability of the model to enable function evaluation, but it also enhances the final accuracy of the symbolic expressions compared to when no function evaluation data is used.

We demonstrate that Tree LSTMs outperform Tree NN's by a large margin with or without function evaluation data in all the experiments. We attribute this to the fact that LSTM cells ameliorate vanishing and exploding gradients along paths in the tree compared to fully-connected blocks used in Tree NNs. This enables the model to be capable of reasoning in equations of higher depth where reasoning is a more difficult task compared to an equation of lower depth. Therefore, it is important to use both a tree and a cell with memory, such as an LSTM cell for modeling the properties of mathematical functions.

**Implementation Details** Our neural networks are developed using MxNet (Chen et al., 2015). All the experiments and models are tuned over the same search space and the reported results are the best achievable prediction accuracy for each method. We use $L2$-regularization as well as dropout to avoid overfitting, and train all the models for 100 epochs. We have tuned for the hidden dimension $\{10,20,50\}$, the optimizers $\{$SGD, NAG (Nesterov accelerated SGD), RMSProp, Adam, AdaGrad, AdaDelta, DCASGD, SGLD (Stochastic Gradient Riemannian Langevin Dynamics)$\}$, dropout rate $\{0.2,0.3\}$, learning rate $\{10^{-3}, 10^{-5}\}$, regularization ratio $\{10^{-4}, 10^{-5}\}$ and momentum $\{0.2,0.7\}$. Most of the networks achieved their best performance using Adam optimizer Kingma & Ba (2014) with a learning rate of 0.001 and a regularization ratio of $10^{-5}$. Hidden dimension and dropout varies under each of the scenarios.

## 4 EXPERIMENTS AND RESULTS

We indicate the complexity of an identity by its depth. We setup the following experiments to evaluate the performance and the generalization capability of our proposed framework. We investigate the

---

[2]Our dataset generation method, proposed model, and data is available here: `https://github.com/ForoughA/neuralMath`

Table 3: **Generalization Results:** the train and the test contain equations of the same depth [1,2,3,4]. Results are on unseen equations. *Sym* refers to accuracy of Symbolic expressions and *F Eval* refers to MSE of function evaluation expressions. The last four columns measure the accuracy of symbolic expressions of different depths.

| Approach | Sym | F Eval | depth 1 | depth 2 | depth 3 | depth 4 |
|---|---|---|---|---|---|---|
| Test set size | 3527 | 401 | 7 | 542 | 2416 | 563 |
| Majority Class | 50.24 | - | 28.57 | 45.75 | 52.85 | 43.69 |
| Sympy | 81.74 | - | 85.71 | 89.11 | 82.98 | 69.44 |
| RNN | 66.37 | - | 57.14 | 62.93 | 65.13 | 72.32 |
| LSTM | 81.71 | - | 85.71 | 79.49 | 80.81 | 83.86 |
| TreeNN | 92.06 | - | **100.0** | 95.37 | 94.16 | 87.45 |
| TreeLSTM | 95.18 | - | 85.71 | 96.50 | 95.07 | 94.50 |
| TreeNN + data | 93.60 | 0.191 | **100.0** | 94.1 | 93.13 | 95.11 |
| TreeLSTM + data | **97.20** | **0.047** | 71.42 | **98.29** | **97.45** | **96.00** |

behavior of the learned model on two different tasks of equation verification and equation completion. Under both tasks, We assess the results of the method on symbolic as well as function evaluation expressions. We compare each of the models with sequential Recurrent Neural Networks (RNN), LSTMs and recursive tree neural networks also knows as Tree NN's. Moreover, we show the effect of adding function evaluation data to the final accuracy on symbolic expressions. All the models train on the same dataset of symbolic expressions. Models, *Tree LSTM + data* and *Tree NN + data* use function evaluation data on top of the symbolic data.

Our dataset consists of 17632 symbolic equations, 8902 of which are correct. This data includes 39 equations of depth 1, 2547 equations of depth 2, 12217 equations of depth 3 and 2836 equations of depth 4. It should be noted that the equations of depth 1 and 2 have been maxed out in the data generation. We also add 2000 function evaluation equations and decimal expansion trees for numbers that includes 1029 correct samples. We have 2 equations of depth 1, 831 equations of depth 2 and 1128 equations of depth 3 in this function evaluation dataset. Our numerical data includes 30% of numbers of precision 2 in the range $[-3.14, 3.14]$ chosen at random.

**Equation Verification - Generalization to Unseen Identities:** In this experiment we randomly split all of the generated data that includes equations of depths 1 to 4 into train and test partitions with an 80%/20% split ratio. We evaluate the accuracy of the predictions on the held-out data. The results of this experiment are presented in Table 3. As it can be seen, tree structured networks are able to make better predictions compared to chain-structured or flat networks. Therefore, we are leveraging the structure of the identities to capture information about their validity. Moreover, the superiority of Tree LSTM to Tree NN shows that it is important to incorporate cells that have memory. The detailed prediction accuracy broken in terms of depth and also in terms of symbolic and function evaluation expressions is also given in Table 3.

**Equation Verification - Extrapolation to Unseen Depths:** Here we evaluate the generalization of the learned model to equations of higher and lower complexity. Generalization to equations of higher depth indicates that the network has been able to learn the properties of each mathematical function and is able to use this in more complex equations to verify their correctness. Ability to generalize to lower complexity indicates whether the model can infer properties of simpler mathematical functions by observing their behavior in complex equations. For each setup, we hold out symbolic expressions of a certain depth and train on the remaining depths. Table 4 presents the results of both setups, which suggest that Tree LSTM trained on a combination of symbolic and function evaluation data, outperforms all other methods across all metrics. Comparing the symbolic accuracy of Tree models with and without the function evaluation data, we conclude that our models are able to utilize the patterns in the function evaluations to improve and better model the symbolic expressions as well.

**Equation Completion:** In this experiment, we evaluate the capability of the model in completing equations by filling in a blank in unseen identities. For this experiment, we use the same models as

Table 4: **Extrapolation Evaluation** to measure capability of the model to generalize to unseen depth on symbolic equations

| Approach | Train depth:1,2,3; Test depth: 4 | | | Train depth:1,3,4; Test depth: 2 | | |
|---|---|---|---|---|---|---|
| | **Accuracy** | **Precision** | **Recall** | **Accuracy** | **Precision** | **Recall** |
| Majority Class | 55.22 | 0 | 0 | 56.21 | 0 | 0 |
| RNN | 65.15 | 68.61 | 75.51 | 71.27 | 82.98 | 43.27 |
| LSTM | 76.40 | 71.62 | 78.35 | 79.31 | 75.27 | 79.31 |
| TreeNN | 88.36 | 87.87 | 85.86 | 92.58 | 89.04 | 94.71 |
| TreeLSTM | 93.27 | 90.20 | 95.33 | 94.78 | 94.15 | 93.90 |
| TreeNN + data | 93.34 | 90.34 | 95.33 | 93.36 | 89.75 | 95.78 |
| TreeLSTM + data | **96.17** | **92.97** | **97.15** | **97.37** | **96.08** | **96.86** |

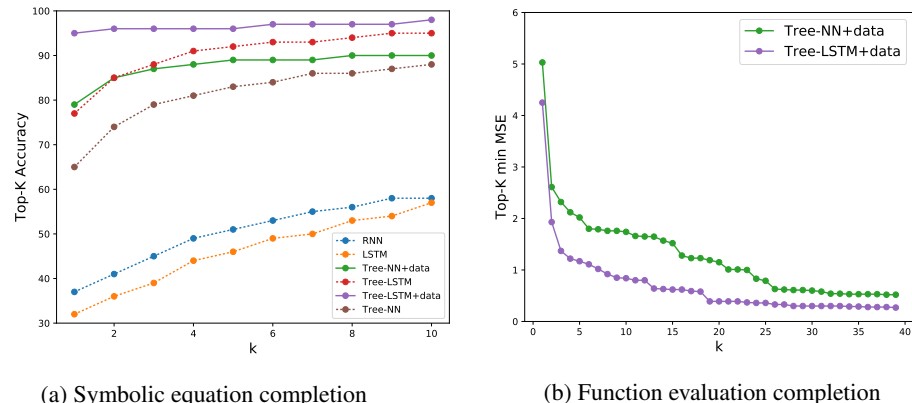

(a) Symbolic equation completion

(b) Function evaluation completion

Figure 3: **Evaluating Equation Completion**, Figure 3a shows the top-$k$ accuracy of the symbolic data for different methods, and Figure 3b illustrates the minimum MSE of the top-k predictions, for the function evaluation data.

reported in table 3. We take all the test equations and randomly choose a node of depth either 1 or 2 in each equation, and replace it with all possible configurations of depth 1 and 2 expressions from our grammar. We then give this set of equations to the models and look at the top-$k$ predictions for the blank ranked by the model's confidence. We perform equation completion on both symbolic and function evaluation expressions.

Figure 3a shows the accuracy of the top-$k$ predictions vs. $k$, for the symbolic expressions. We define the top-$k$ accuracy as the percentage of samples for which there is at least one correct match for the blank in the top $k$ predictions. This indicates that the hardest task is to have a high accuracy for $k = 1$. Therefore, as in Fig 3a, the differences at $k = 1$ for models that use function evaluation data vs. models that do not, indicates the importance of combining symbolic and function evaluation data for the task of equation completion. We can also see that tree-structured models are substantially better than sequential models, indicating that it is important to capture the compositionality of the data in the structure of the model. Finally, Tree LSTM shows superior performance compared to Tree NN under both scenarios.

Figure 3b evaluates equation completion on function evaluation expressions by measuring the top-$k$ minimum MSE for different $k$s. We define the top-$k$ minimum MSE as the MSE between the true value of the blank and the closest prediction to the true value among the top-$k$ predictions. Similar to the top-$k$ accuracy, the hardest task is to have a low MSE for $k = 1$ since it indicates that the correct prediction is the first model prediction. We would like to note that, for function evaluation expressions, there is only one correct prediction for a blank, whereas for symbolic expressions, there may be many correct candidates for a specific blank. This evaluation is performed only for models

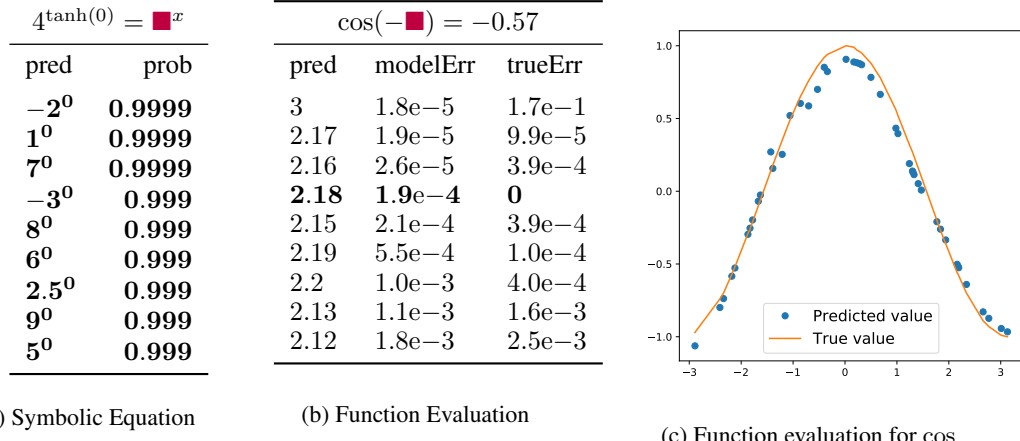

|  |  |
| --- | --- |
| $4^{\tanh(0)} = \blacksquare^x$ | |

| pred | prob |
| --- | --- |
| $-2^0$ | **0.9999** |
| $1^0$ | **0.9999** |
| $7^0$ | **0.9999** |
| $-3^0$ | **0.999** |
| $8^0$ | **0.999** |
| $6^0$ | **0.999** |
| $2.5^0$ | **0.999** |
| $9^0$ | **0.999** |
| $5^0$ | **0.999** |

| $\cos(-\blacksquare) = -0.57$ | | |
| --- | --- | --- |
| pred | modelErr | trueErr |
| 3 | 1.8e−5 | 1.7e−1 |
| 2.17 | 1.9e−5 | 9.9e−5 |
| 2.16 | 2.6e−5 | 3.9e−4 |
| **2.18** | **1.9e−4** | **0** |
| 2.15 | 2.1e−4 | 3.9e−4 |
| 2.19 | 5.5e−4 | 1.0e−4 |
| 2.2 | 1.0e−3 | 4.0e−4 |
| 2.13 | 1.1e−3 | 1.6e−3 |
| 2.12 | 1.8e−3 | 2.5e−3 |

(a) Symbolic Equation

(b) Function Evaluation

(c) Function evaluation for cos

Figure 4: **Examples of Equation Completion** of the Tree LSTM + data model. Figures 4a and 4b show examples of equation completion from the test set where the predictions are ranked by model's confidence and the correct prediction is shown in boldface. Figure 4c depicts the predicted values of cos(x) with blue dots for x in [-3.14,3.14] in the test set

that use function evaluation data. We can see from the figure, that Tree LSTM's MSE is better that that of Tree NN across all $k$s.

We present examples of equations and generated candidates in Figures 4a and 4b for the Tree LSTM + data model. Figure 4a presents the results on a symbolic equation for which the correct prediction value is 1. The Tree LSTM is able to generate many candidates with a high confidence, all of which are correct. Column *prob* in the figure is the output probability of softmax which indicates the model's confidence in its prediction. On the other hand, in Figure 4b, we show a function evaluation example, where the correct answer is 2.18 rounded to precision 2. The correct answer is among the top predictions as shown in Figure 4b. All the predicted values for the blank are listed in column *pred* ranked by the model's prediction confidence. Column *modelErr* shows the model's confidence of prediction, which is the squared error between the predicted value of cos(-pred) and $-0.57$. Column *trueErr* is the squared error between the true value of cos(-pred) rounded to precision 2 and $-0.57$. As it is shown, the predicted candidates are close to the true value. It is worth noting that for the function evaluation task of Figure 4b, there is only 1 correct answer, whereas for the task in Figure 4a there can be many correct solutions. We also present example predictions of our model for function evaluations by plotting the top predicted values for cos on samples of test data in Figure 4c.

## 5 Conclusions and Future Work

In this paper we proposed combining black-box function evaluation data with symbolic expressions to improve the accuracy and broaden the applicability of previously proposed models in this domain. We apply this to the novel task of validating and completing mathematical equations. We studied the space of trigonometry and elementary algebra as a case study to validate the proposed model. We also proposed a novel approach for generating a dataset of mathematical identities and generated identities in trigonometry and elementary algebra. As noted, our data generation technique is not limited to trigonometry and elementary algebra. We show that under various experimental setups, Tree LSTMs trained on a combination of symbolic expressions and black-box function evaluation achieves superior results compared to the state-of-the-art models.

In our future work we will expand our testbed to include other mathematical domains, inequalities, and systems of equations. What is interesting about multiple domains is to investigate if the learned representations for one domain can be transfered to the other domains, and whether the embedding of each domain are clustered close to each other similar to the way the word embedding vectors behave. We are also interested in exploring recent neural models with addressable differentiable memory, in order to evaluate whether they can handle equations of much higher complexity.

ACKNOWLEDGMENTS

The authors would like to thank Amazon Inc., for the AWS credits. F. Arabshahi is supported by DARPA Award D17AP00002. A. Anandkumar is supported by Microsoft Faculty Fellowship, NSF CAREER Award CCF-1254106, DARPA Award D17AP00002 and Air Force Award FA9550-15-1-0221. S. Singh would like to thank Adobe Research and FICO for supporting this research.

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
