# OpenReview forum: "Combining Symbolic Expressions and Black-box Function Evaluations in Neural Programs"
_ICLR.cc/2018/Conference — Accept (Poster)_

### Official Review · AnonReviewer3 · 2017-11-24
**A dataset paper with application of TreeLSTMs**

**Rating:** 6
**Confidence:** 4

**Review:**

Summary

This paper presents a dataset of mathematical equations and applies TreeLSTMs to two tasks: verifying and completing mathematical equations. For these tasks, TreeLSTMs outperform TreeNNs and RNNs. In my opinion, the main contribution of this paper is this potentially useful dataset, as well as an interesting way of representing fixed-precision floats. However, the application of TreeNNs and TreeLSTMs is rather straight-forward, so in my (subjective) view there are only a few insights salvageable for the ICLR community and compared to Allamanis et al. (2017) this paper is a rather incremental extension.

Strengths

The authors present a new datasets for mathematical identities. The method for generating additional correct identities could be useful for future research in this area.
I find the representation of fixed-precision floats presented in this paper intriguing. I believe this contribution should be emphasized more as it allows the model to generalize to unseen numbers and I am wondering whether the authors see some wider application of this representation for neural programming models.
I liked the categorization of the related work.

Weaknesses

p2: It is mentioned that the framework is the first to combine symbolic expressions with black-box function evaluations, but I would argue that Neural Programmer-Interpreters (NPI; Reed & De Freitas) are already doing that (see Fig 1 in that paper where the execution trace is a symbolic expression and some expressions "Act(LEFT)" are black-box function applications directly changing the image).
The differences to Allamanis et al. (2017) are not worked out well. For instance, the authors use the TreeNN model from that paper as a baseline but the EqNet model is not mentioned at all. The obvious question is whether EqNets can be applied to the two tasks (verifying and completing mathematical equations) and if so why this has not been done.
The contribution regarding black box function application is unclear to me. On page 6, it is unclear to me what "handles […] function evaluation expressions". As far as I understand, the TreeLSTM learns to the return value of function evaluation expressions in order to predict equality of equations, but this should be clarified.
I find the connection of the proposed model and task to "neural programming" weak. For instance, as far as I understand there is no support for stateful programs. Furthermore, it would be interesting to hear how this work can be applied to existing programming languages such as Haskell. What are the limitations of the architecture? Could it learn to identify equality of two lists in Haskell?
p6: The paragraph on baseline models is rather uninformative. TreeLSTMs have been shown to outperform Tree NN's in various prior work. The statement that "LSTM cell […] helps the model to have a better understanding of the underlying functions in the domain" is vague. LSTM cells compared to fully-connected layers in Tree NNs ameliorate vanishing and exploding gradients along paths in the tree. Furthermore, I would like to see a qualitative analysis of the reasoning capabilities that are mentioned here. Did you observe any systematic differences in the ~4% of equations where the TreeLSTM fails to generalize (Table 3; first column).

Minor Comments

Abstract: "Our framework generalizes significantly better" I think it would be good to already mention in comparison to what this statement is.
p1: "aim to solve tasks such as learn mathematical" -> "aim to solve tasks such as learning mathematical"
p2: You could add a citation for Theano, Tensorflow and Mxnet.
p2: Could you elaborate how equation completion is used in Mathematical Q&A?
p3: Could you expand on "mathematical equation verification and completion […] has broader applicability" by maybe giving some concrete examples.
p3 Eq. 5: What precision do you consider? Two digits?
p3: "division because that they can" -> "division because they can"
p4 Fig. 1: Is there a reason 1 is represented as 10^0 here? Do you need the distinction between 1 (the integer) and 1.0 (the float)?
p5: "we include set of changes" -> "we include the set of changes"
p5: In my view there is enough space to move appendix A to section 2. In addition, it would be great to see more examples of generated identities at this stage (including negative ones).
p5: "We generate all possible equations (with high probability)" – what is probabilistic about this?
p5: I don't understand why function evaluation results in identities of depth 2 and 3. Is it both or one of them?
p6: The modules "symbol" and "number" are not shown in the figure. I assume they refer to projections using Wsymb and Wnum?
p6: "tree structures neural networks" -> "tree structured neural networks"
p6: A reference for the ADAM optimizer should be added.
p6: Which method was used for optimizing these hyperparameters? If a grid search was used, what intervals were used?
p7: "the superiority of Tree LSTM to Tree NN shows that is important to incorporate cells that have memory" is not a novel insight.
p8: When you mention "you give this set of equations to the models look at the top k predictions" I assume you ranked the substituted equations by the probability that the respective model assigns to it?
p8: Do you have an intuition why prediction function evaluations for "cos" seem to plateau certain points? Furthermore, it would be interesting to see what effect the choice of non-linearity on the output of the TreeLSTM has on how accurately it can learn to evaluate functions. For instance, one could replace the tanh with cos and might expect that the model has now an easy time to learn to evaluate cos(x).
p8 Fig 4b; p9: Relating to the question regarding plateaus in the function evaluation: "in Figure 4b […] the top prediction (0.28) is the correct value for tan with precision 2, but even other predictions are quite close" – they are all the same and this bad, right?
p9: "of the state-of-the-art neural reasoning systems" is very broad and in my opinion misleading too. First, there are other reasoning tasks (machine reading/Q&A, Visual Q&A, knowledge base inference etc.) too and it is not obvious how ideas from this paper translate to these domains. Second, for other tasks TreeLSTMs are likely not state-of-the-art (see for example models on the SQuAD leaderboard: https://rajpurkar.github.io/SQuAD-explorer/) .
p9: "exploring recent neural models that explicitly use memory cells" – I think what you mean is models with addressable differentiable memory.

# Update after the rebuttal
Thank you for the in-depth response and clarifications. I am increasing my score by one point. I have looked at the revised paper and I strongly suggest that you add the clarifications and in particular comments regarding comparison to related work (NPI, EqNet etc) to the paper. Regarding Fig. 4b, I am still not sure why all scores are the same (0.9977) -- I assume this is not the desired behavior?

---

> ### Author Response · Authors · 2018-01-05
> **Response to reviewer 3 - part 1**
>
> We would like to thank the reviewer for the constructive feedback. Here is our response:
>
> Novelty: We would like to emphasize that the main contribution of the paper is combining high-level symbolic and function evaluation expressions, which none of the existing work has done. We are proposing a new framework for modeling mathematical equations. This framework includes defining new problems, equation validation and equation completion, as well as introducing a dataset generation method and a recursive neural network that combines these function evaluation and symbolic expressions. Indeed treeLSTMs are not new, however, using them to combine both the symbolic and function evaluation expressions by incorporating different loss functions and terminal types is novel.
>
> I believe this contribution [decimal expansion of numbers] should be emphasized more: We have emphasized this contribution as explained in the revisions.
>
> Neural Programmer-Interpreters (NPI; Reed & De Freitas) are already combining symbolic expressions and function evaluations: We would like to point out that NPI is using execution traces which are hard to obtain compared to symbolic expressions. Symbolic expressions can be thought of as the computation graph of a program. Moreover, we do not ground our symbolic expressions at a specific node in the tree which is the case in Figure 1 of the NPI paper. Rather, we have high-level symbolic expressions that summarize the behavior of mathematical functions and apply to many groundings of each formula. These symbolic expressions are combined with input-output examples of the functions such as sin(-2.5)=-0.6. This not only helps the final model’s accuracy, but also enables applications such as equation completion.
>
> Comparison to EqNets: Thank you for pointing this out. Have made the destintion more clear in the baseline section as explained in the revisions. We would like to point that EqNets are designed for finding equivalence classes in a symbolic dataset, whereas we are aiming to verify math identities. In our dataset of trigonometric identities and algebraic equations we have way too many equivalence classes. Moreover, in our setting with function evaluation expressions we have so many classes that it does not make sense to use the EqNet’s approach to the problem. We argue that our framework is better since it allows equation completion, which is not possible in EqNets.
>
> What do we mean by handling both symbolic and function evaluation expression: What we mean is that our network architecture accounts for symbolic terminals using one-hot encoding vectors input to a 1 layer neural network whereas it accounts for floating point numbers for function evaluations through a 2layer neural network that takes as its input the literal value of the floating point number. We have also added an auto-encoder and changed the loss function of the function evaluation training to MSE in order to improve the results of the function evaluation curve and resolve the plateau of values in Figure 4(c). These changes are explained in the revision description. Please take a look at the updated version of Figure 4(c) in the revised paper.
>
> Application to neural programming: The connection to neural programs is the fact that our tree structure reflects the expression structure of a program. Applying this to Haskell-like and
> stateful programs would be a good revenue for future work.
>
> LSTM cell […] helps the model to have a better understanding of the underlying functions in the domain: We have changed this in the paper
>
> response continues in the next post...

---

> > ### Author Response · Authors · 2018-01-05
> > **Response to reviewer 3 - part 2**
> >
> > Typos: fixed
> >
> > Theano, … Citations: added
> >
> > Use of equation completion in Q/A: For example in a math Q/A system if someone wants to know the value of sin^2(\theta)+cos^2(\theta) we are able to answer that the result is 1. This is not possible for EqNets.
> >
> > Concrete example of the broader applicability of equation completion and equation verification. By this we mean that our network can always identify equivalence classes, since it validates the equality of two sides of the equations. Moreover, it enables equation completion which is not possible for the frameworks proposed by (Allamanis et al 2016) and (Zaremba et al 2014). For example we can verify that 2*x^2 and x^2+x^2 are equivalent which is the task studied by both Allamanis et al and Zaremba et al. Moreover we can validate correctness of classes that have never been seen in the training dataset since we are proposing a new application which is equation validation. EqNets and similar models rely on classification and therefore have difficulty generalizing to classes unseen in the training data, but we are able to validate identities whose equivalence class has never been observed in the training set. Moreover, Allamanis et. al. and Zaremba et al are not able to complete a given equation such as 2*x^2 = x^2 + __. Whereas our algorithm can do this enabling a broader applicability.
> >
> > Precision: Precision is two. This is fixed in the paper as described in the revisions.
> >
> > Difference between 1.0 and 1: We are distinguishing this to make sure that the network learns that the vector representation of 1.0, inputed as a literal, is equivalent to the vector representation of 1, inputed as a one-hot encoding, and is equivalent to its decimal representation 10^0. We already have other equations such as x^0=1 and 1.0=10^0 in our training set. This is our novelty to understand the decimal encoding.
> >
> > Add details of data generation back to the paper: Done,
> >
> > Add examples of generated equations: This is added to the paper and can be seen in Table 2
> >
> > What is probabilistic about generating all possible equations: We removed the high probability claim
> >
> > Depth of Function evaluation equations: This generated both equations of depth 2 and 3. For example, sin(2.5)=0.6 is an equation of depth 2 and sin(-2.5)=-0.6 is of depth 3 (look at Figure 2(c)).
> >
> > Symb and Num modules not shown in Fig 1: Fig 1 shows the actual parse tree of the input equations and not the modules of the neural network. We will later define the w_sym and w_num module in the structure of the neural network depending on the terminal in each input equation.
> >
> > Adam citation:  Added
> >
> > Optimization method: Grid search. This is added to the paper as explained in the revisions.
> >
> > the superiority of Tree LSTM to Tree NN shows that is important to incorporate cells that have memory: Paper modified accordingly
> >
> > ranked the substituted equations for equation completion: This is correct, we rank equations for equation completion using the probability that the network assigns to it at the softmax layer output.
> >
> > evaluations for "cos" seem to plateau: We have corrected this curve by changing the training loss of function evaluation equation to MSE and training an auto-encoder that encodes the value of the floating point numbers and then outputs a number for a given vector embedding. This is explained in the paper revision.
> >
> > All the values of Figure 4(b) are close and this is bad: This was also resolved with the MSE training idea
> >
> > exploring recent neural models that explicitly use memory cells means models with addressable differentiable memory: Yes thank you for pointing this we have corrected this in the paper
> >
> > of the state-of-the-art neural reasoning systems is too broad: This is fixed in the paper

---

> ### Author Response · Authors · 2018-01-26
> **Response to reviewer 3, after the rebuttal**
>
> Thank you for taking the time to go through our response. We are pleased to know that we were able to address your concerns and make clarifications. We have uploaded a revision in which we have included the clarifications regarding NPI. Page 7, Paragraph 3 Includes clarification about comparison with EqNets. Thank you for pointing this out. We have updated Figure 4(b) with the results of MSE training. The output probabilities are now replaced with the prediction squared error as shown in the revised version. We have written the complete list of revisions in the *List of Revisions* post **update after rebuttal** section.

---

### Official Review · AnonReviewer2 · 2017-11-25
**Impressive model for learning to encode symbols and numbers, and to evaluate functions, for determining the validity of algebraic/trigonometric equalities**

**Rating:** 8
**Confidence:** 3

**Review:**

SUMMARY

The model evaluates symbolic algebraic/trigonometric equalities for validity, with an output unit for validity level at the root of a tree of LSTM nodes feeding up to the root; the structure of the tree matches the parse tree of the input equation and the type of LSTM cell at each node matches the symbol at that node in the equation: there is a different cell type for each symbol. It is these cell types that are learned. The training data includes labeled true and false algebraic/trigonometric identities (stated over symbols for variables) as well as function-evaluation equalities such as "tan(0.28) = 0.29" and decimal-expansion equations like "0.29 = 2*10^(-1) + 9*10^(-2)".  I believe continuous values like "0.29" in the preceding expressions are encoded as the literal value of a single unit (feeding into an embedding unit of type W_{num}), whereas the symbols proper (including digit numerals) are encoded as 1-hot vectors (feeding into an embedding unit of type W_{symb}).
Performance is at least 97% when testing on unseen expressions of the same depth (up to 4) as the training data. Performance when trained on 3 levels (among 1 - 4) and testing on generalization to the held-out level is at least 96% when level 2 is held out, at least 92% when level 4 is withheld. Performance degrades (even on symbolic identities) when the function-evaluation equalities are omitted, and degrades when LSTM cells are replaced by plain RNN cells. The largest degradation is when the tree structure is replaced (presumably) by a sequence structure.
Performance was also tested on a fill-in-the-blank test, where a symbol from a correct equation was removed and all possible replacements for that symbol with expressions of depth up to 2 were tested, then ranked by the resulting validity score from the model. From the graph it looks like an accuracy of about 95% was achieved for the 1-best substituted expression (accuracy was about 32% for a sequential LSTM).

WEAKNESSES

* The title is misleading; "blackbox function evaluation" does not suggest what is intended, which is training on function-evaluation equations. The actual work is more interesting than what the title suggests.
* The biggest performance boost (roughly 15%) arises from use of the tree structure, which is given by an oracle (implemented in a symbolic expression parser, presumably): the network does not design its own example-dependent structure.
* What does the sympy baseline mean in Table 2? We are only told that sympy is a "symbolic solver". Yet the sympy performance scores are in the 70-80% range. If the solver’s performance is that weak, why is it used during generation of training data to determine the validity of possible equations?
* Given that this is a conference on "learning representations" it would have been nice to see at least a *little* examination of the learned representations. It would be easy to do some interesting tests. How well does the vector embedding for "2*10^(-1) + 9*10^(-2)" match the vector for the real value 0.29? W_{num} embeds a continuum of real values in R^d: what is this 1-dimensional embedding manifold like? How do the embeddings of different integers provided by W_{sym} relate to one another? My rating would have been higher had there been some analysis of the learned representations.
* We are told only that the "hidden dimension … varies"; it would be nice if the text or results tables gave at least some idea of what magnitude of embedding dimension we’re talking about.

STRENGTHS

The weaknesses above notwithstanding, this is a very interesting piece of work with impressive results.
* The number of functions learned, 28, is a quantum jump from previous studies using 8 or fewer functions.
* It is good to see the power of training the same system to learn the semantics of functions from the equations they satisfy AND from the values they produce.
* The inclusion of decimal-expansion equations for relating numeral embeddings to number embeddings is clever.
* The general method used for randomly generating a non-negligible proportion of true equations is useful.
* The evaluation of the model is thorough and clear.
* In fact the exposition in the paper as a whole is very clear.

---

> ### Author Response · Authors · 2018-01-05
> **Response to reviewer 2**
>
> We would like to thank the reviewer for the constructive feedback. Here is our response:
>
>  I believe continuous values like "0.29" in the preceding expressions are encoded as the literal value of a single unit (feeding into an embedding unit of type W_{num}), whereas the symbols proper (including digit numerals) are encoded as 1-hot vectors (feeding into an embedding unit of type W_{symb}).
> This understanding is correct
>
> Misleading title: We can change the title to “Combining Symbolic and Function Evaluation Expressions for Training Neural Programs” if the reviewer feels that this reflects better what we are doing.
>
> Example-dependent structure: If I understand the question correctly, the network’s structure is indeed example dependent and is indicated by the input equation. We have not used any symbolic expression parser to construct the equation parses. The neural network’s structure is dynamic and its structure depends on the parse of the input equation that comes naturally with it. We hypothesize that the input equation’s expression tree in indeed the best compositionality one can obtain as it represents the natural composition of the equations. In fact, if we had access to this clear composition tree in NLP tasks, the models would have been more accurate. many programming languages have access to the tree structure of the program. Without the tree, the problem will be very challenging and we would investigate learning the structure in the future.
>
> Sympy performance:
> We used Sympy to check the correctness of the generated equations. If the correctness of an equation is verified by sympy then it is added to the dataset. Therefore, sympy has a 100% accuracy for predicting correct equalities in our dataset. It is only the incorrect equalities that cause Sympy’s performance to drop as we explain below.
> In order to assess Sympy’s performance, we give each equation to Sympy. It either returns, True, or False, or returns the equation in its original form (indicating that sympy is incapable of deciding whether the equality holds or not). Let’s call this the Unsure class. In the reported Sympy accuracies we have treated the Unsure class as a miss-classification. Another approach is to perform majority class prediction on the Unsure class. This will result in the same number as shown in table 2 since our majority class is True (50.24% are correct). In order to be fair, we can also treat the Unsure class as a fair coin toss and report half as correctly predicted. If we do this, the Sympy row in table 2 will be updated with these numbers: 90.81 & - & 90.00 & 94.46 & 91.45 & 84.54. If the reviewer believes that this approach is better we can update these numbers in Table 2.
> We have also added this explanation to the paper. Given some other solver as oracle for adding equations, it would have been interesting to evaluate the accuracy of sympy for predicting the correctness of those equations.
>
> Examination of learned representations: This is a very good suggestion. In order to examine the learned representations we have depicted Figure 4. In order to see how close the vector embedding of an expression, say cos(0.8) is to the vector embedding of 0.69, we have presented Figure 4(c) which is similar to what the reviewer is suggesting about the decimal representation, if we understand correctly. We have also performed a minor modification as explained in the revisions, which makes it easier to interpret the learned representation. Our W_num block encodes the representation of floating point numbers like 0.29. We have trained a decoder, W_num^{-1}, that decodes the d-dimensional representation of 0.29 back to the actual number. Moreover, Table 4(b) indicates that the vector embeddings of tan(x) for x close to 0.28 results in vector embeddings that are close to 0.29 which is the correct value for tan(0.28) with precision 2.
>
> Hidden dimension: The hidden dimension is chosen from set {10, 20, 50}. This has been added to the paper as explained in the revisions.

---

### Official Review · AnonReviewer1 · 2017-11-29
**A review of "Combining  Symbolic Expressions..."**

**Rating:** 5
**Confidence:** 4

**Review:**

This paper proposes a model that predicts the validity of a mathematical expression (containing trigonometric or elementary algebraic expressions) using a recursive neural network (TreeLSTM).  The idea is to take the parse tree of the expression, which is converted to the recursive neural network architecture, where weights are tied to the function or symbol used at that node.  Evaluation is performed on a dataset generated specifically for this paper.

The overall approach described in this paper is technically sound and there are probably some applications (for example in online education).  However the novelty factor of this paper is fairly low — recursive neural nets have been applied to code/equations before in similar models.  See, for example, “Learning program embeddings to propagate feedback on student code” by Piech et al, which propose a somewhat more complex model applied to abstract syntax trees of student written code.

I’m also not completely sure what to make of the experimental results.  One weird thing is that the performance does not seem to drop off for the models as depth grows.  Another strange thing is that the accuracies reported do not seem to divide the reported test set sizes (see, e.g., the depth 1 row in Table 2).  It would also be good to discuss the Sympy baseline a bit — being symbolic, my original impression was that it would be perfect all the time (if slow), but that doesn’t seem to be the case, so some explanation about what exactly was done here would help.  For the extrapolation evaluation — evaluating on deeper expressions than were in the training set — I would have liked the authors to be more ambitious and see how deep they could go (given, say, up to depth 3 training equations).

---

> ### Author Response · Authors · 2018-01-05
> **Response to reviewer 1**
>
> We would like to thank the reviewer for the constructive feedback. Here is our response:
>
> the novelty factor of this paper is fairly low: We would like to emphasize that the main contribution of the paper is combining high-level symbolic and function evaluation expressions, which none of the existing work has done. We are proposing a new framework for modeling mathematical equations. This framework includes defining new problems, equation validation and equation completion, as well as introducing a dataset generation method and a recursive neural network that combines these function evaluation and symbolic expressions. Indeed treeLSTMs are not new, however, using them to combine both the symbolic and function evaluation expressions by incorporating different loss functions and terminal types is novel.
>
> Dataset only for this paper: This dataset makes sure that there is a good coverage of the properties of different mathematical functions which is critical for proper training of the model. It contains correct and incorrect math equations. An alternative approach that extracts equations online, unfortunately, does not result in enough equations for training. Moreover, we cannot obtain any negative equations. Our proposed dataset generation results in a large scale data of any mathematical domain given a small number of axioms from that domain.
>
> Performance drop:
> One weird thing is that the performance does not seem to drop off for the models as depth grows
> -- It is indeed interesting that the performance drops only marginally as depth grows in Table 2. The reason is, In this experiment we are assessing generalizability to unseen equations and not unseen depths. Therefore, the training data has access to equations of all depths. The results indicate that the model has learned to predict equations of all depths well. It also indicates that the dataset has a good coverage of equations of all depths that ensures good performance across all depths.
> -- Furthermore, the extrapolation experiment (table 3) shows that generalization to equations of smaller depth is easier than generalization to higher depth equations.
> Another strange thing is that the accuracies reported do not seem to divide the reported test set sizes (see, e.g., the depth 1 row in Table 2)
> There was a typo in table 2 in row “test set size” and column “depth 1”, which we have addressed in the revision. The correct number is 5+3 instead of 5+2. Thank you for pointing this out.
>
> Sympy performance:
> --We used Sympy to check the correctness of the generated equations. If the correctness of an equation is verified by sympy then it is added to the dataset. Therefore, sympy has a 100% accuracy for predicting correct equalities in our dataset. It is only the incorrect equalities that cause Sympy’s performance to drop as we explain below.
> --In order to assess Sympy’s performance, we give each equation to Sympy. It either returns, True, or False, or returns the equation in its original form (indicating that sympy is incapable of deciding whether the equality holds or not). Let’s call this the Unsure class. In the reported Sympy accuracies we have treated the Unsure class as a miss-classification. Another approach is to perform majority class prediction on the Unsure class. This will result in the same number as shown in table 2 since our majority class is True (50.24% are correct). In order to be fair, we can also treat the Unsure class as a fair coin toss and report half as correctly predicted. If we do this, the Sympy row in table 2 will be updated with these numbers: 90.81 & - & 90.00 & 94.46 & 91.45 & 84.54. If the reviewer believes that this approach is better we can update these numbers in Table 2.
> --We have also added this explanation to the paper. Given some other solver as oracle for adding equations, it would have been interesting to evaluate the accuracy of sympy for predicting the correctness of those equations.
>
> Accuracy on equations of larger depth given up to depth 3 equations: This is indeed a very interesting test. We performed this test for training equations of up to depth 3 and the performance on equations of depth 5 is 89% and for equations of depth 6 is 85% for the treeLSTM model + data. We will add the full results of this extra experiment to the paper.

---

### Author Response · Authors · 2018-01-06
**List of revisions:**


**Reviewer 1:**

Page 8, Table 3: fixed typo for the number of equations in the test set for depth 1 column

Page 7, parag 1: we have included the explanation about the Sympy baseline as suggested by both reviewers 1 and 2.

References: Added Piech et al 2015

**Reviewer 2:**

page 6, paragraph 2 described that the input parse of the equation is inherent within each equation.

Page 6, last subsection: We have modified the model to correct Figure 4(c) as explained in this sections. This also indicated how our model handles both function evaluation and symbolic expressions, which was a question raised by reviewer 3. In this modification we have replaced the logistic loss, with the MSE loss for function evaluation expressions. We have also added an autoencoder design for embedding numbers in order to have a better understanding of the learned representations as suggested by reviewer 2.

Page 7, parag 1: we have included the explanation about the Sympy baseline as suggested by both reviewers 1 and 2.

Page 7, implementation details: We have included the search space for tuning as suggested by reviewers 2 and 3

**Reviewer 3:**

Page 2, parag 1: In order to address reviewer 3’s comment we have added citations for Theano and tensorFlow.

Page 2, Parag 3: emphasized the importance of the decimal expansion tree as suggested by reviewer 3.

Page 3, equation 5: added the precision to address reviewer 3’s comment

Page 3, last parag: fixed typos mentioned by reviewer 3

Page 5, Parag 2: Included the details for data generation as suggested by reviewer 3

Page 5, parag 6: removed the high probability claim as suggested by reviewer 3

Page 5, parag 6: Added text that explains Table 2,

Page 6: added Table 2,  including examples of generated equations as suggested by reviewer 3

Page 6, last subsection: We have modified the model to correct Figure 4(c) as explained in this sections. This also indicated how our model handles both function evaluation and symbolic expressions, which was a question raised by reviewer 3. In this modification we have replaced the logistic loss, with the MSE loss for function evaluation expressions. We have also added an autoencoder design for embedding numbers in order to have a better understanding of the learned representations as suggested by reviewer 2.

Page 7, paragraph 3: We have included reviewer 3’s comment about vanishing and exploding gradients.

Page 7, implementation details: We have included the search space for tuning as suggested by reviewers 2 and 3

Page 9, Figure 4(c): We have updated this curve with the MSE training results to address reviewer 3’s comments about the plateau in the curve in the original submission

Page 10, Conclusions: We have removed the claim about the state-of-the-art neural models and added addressable differentiable memory as suggested by reviewer 3.

References: Added references for Theano, TensorFlow and Adam optimizer.

**Other modifications:**

Page 5: modified Figure 2(b) to incorporate the autoencoder (W_num and W_num^{-1}) and indicate that we are performing MSE training for function evaluation expressions. This change resulted in a correction to Figure 4(c) such that the values do not plateau.

**update after the rebuttal:**
Page 1: title changed
Page 3, Paragraph 1: Added clarification about NPI, pointed by reviewer 3.
Page 9, Paragraph 2: Added description about Figure 4(b).
Page 10, Figure 4(b) is updated with the MSE training results.

---

### Decision · Program_Chairs · 2018-01-29
**ICLR 2018 Conference Acceptance Decision**

**Decision:**

Accept (Poster)

**Comment:**

Learn to complete an equation by filling the blank with a missing function or numeral, and also to evaluate an expression.  Along the way learn to determine if an identity holds (e.g. sin^2(x) + cos^2(x) = 1).  They use a TreeNN with a separate node for each expression in the grammar.

PROS:
1. They've put together a new dataset of equational expressions for learning to complete an equation by filling in the blank of a missing function (or value) and function evaluation.   They've done this in a nice way with a generator and will release it.

2. They've got two interesting ideas here and they seem to work.  First, they train the network to jointly learn to manipulate symbols and to evaluate them.  This helps ground the symbolic manipulations in the validity of their evaluations.  They do this by using a common tree net for both processes with both a symbol node and a number node.  They train on identities (sin^2(x) + cos^2(x) = 1) and also on ground expressions (+(1,2) = 3).  The second idea is to help the system learn the interpretation map for the numerals like the "2" in "cos^2(x) with the actual number 2.  They do this by including equations which relate decimals with their base 10 expansion.  For example 2.5 = 2*10^0 + 5*10^-1.  The "2.5" is (I think) treated as a number and handled by the number node in the network.  The RHS leaves are treated as symbols and handled by the symbol node of the network. This lets them learn to represent decimals using just the 10 digits in their grammar and ties the interpretation of the symbols to what is required for a correct evaluation (in terms of their model this means "aligning" the node for symbol with the node for number).

3. Results are good over what seem to us reasonable baselines

CONS:

1. The architecture isn't new and the idea of representing expression trees in a hierarchical network isn't new either.

2. The writing, to me, is a bit unclear in places and I think they still have some work to do follow the reviewers' advice in this area.

I really wrestled with this one, and I appreciate the arguments that say it's not novel enough but I feel that there is something interesting in here and if the authors do a clean-up before final submission it will be ok.